# Topical Application of Bio-Pulsed Avian MSC-Derived Extracellular Vesicles Enhances Hair Regrowth and Skin Rejuvenation: Evidence from Clinical Evaluation and miRNA Profiling

**DOI:** 10.3390/cimb47070539

**Published:** 2025-07-11

**Authors:** Ju-Sheng Shieh, Yu-Tang Chin, Tsu-Te Yeh, Jiong Jiong Guo, Fung-Wei Chang, Hui-Rong Cheng, Hung-Han Hsu, Wei-Lun Huang, Han-Hsiang Huang, Ya-Yu Hsieh, Chien-Ping Chiang, Shih-Ching Wang

**Affiliations:** 1Department of Periodontology, School of Dentistry, Tri-Service General Hospital, National Defense Medical Center, Taipei 114202, Taiwan; richard_shieh614@163.com (J.-S.S.); yutangchin@gmail.com (Y.-T.C.); candyjung@mail2000.com.tw (H.-R.C.); 2Department of Orthopaedic Surgery, Tri-Service General Hospital, National Defense Medical Center, Taipei 114202, Taiwan; tsutey@gmail.com; 3Medical 3D Printing Center, Tri-Service General Hospital, National Defense Medical Center, Taipei 114202, Taiwan; 4Department of Orthopedics and Sports Medicine, The First Affiliated Hospital of Soochow University, Suzhou 215006, China; drjjguo@163.com (J.J.G.); hunghan@ambiotech.asia (H.-H.H.); 5MOE China-Europe Sports Medicine Belt and Road Joint Laboratory, Soochow University, Suzhou 215006, China; 6Department of Obstetrics and Gynecology, Tri-Service General Hospital, National Defense Medical Center, Taipei 114202, Taiwan; doc30666@gmail.com; 7Department of Research and Development, Ascension Medical Biotechnology Co., Ltd., Taipei 114731, Taiwan; rodrickhuang@ambiotech.asia (W.-L.H.); arielhuang@ambiotech.asia (H.-H.H.); 8Department and Graduate Institute of Pharmacology, College of Medicine, National Taiwan University, Taipei 110225, Taiwan; iamenyala@gmail.com; 9Department of Dermatology, Tri-Service General Hospital, National Defense Medical Center, Taipei 114202, Taiwan; 10Department of Dermatology, Saint Paul’s Hospital, Taoyuan 330049, Taiwan

**Keywords:** extracellular vesicles, Bio-Pulsed priming, avian mesenchymal stem cells, regenerative dermatology, hair follicle activation, skin rejuvenation, clinical trial, microRNA profiling, non-invasive therapy, collagen biosynthesis

## Abstract

Small extracellular vesicles (sEVs) derived from mesenchymal stem cells have emerged as promising therapeutic agents in regenerative dermatology. This study evaluated the safety and efficacy of Bio-Pulsed avian mesenchymal stem cell-derived sEVs (AMSC-sEVs), topically applied for hair follicle stimulation and skin rejuvenation. Two prospective, single-arm clinical trials were conducted: one involving 30 participants using a hair ampoule over 60 days, and the other involving 30 participants applying a facial essence for 28 days. Objective measurements demonstrated significant improvements in the anagen/telogen hair ratio, reduced shedding, increased collagen density, and reduced wrinkle depth and pigmentation. Small RNA sequencing and qPCR profiling confirmed that Bio-Pulsed AMSC-sEVs were enriched with regenerative microRNAs, such as miR-21-5p and miR-199a-5p, associated with anti-inflammatory and anti-aging effects. No adverse events were reported. These findings suggest that Bio-Pulsed AMSC-sEVs may offer a safe, non-invasive, and cell-free approach to enhance skin and hair regeneration in human subjects.

## 1. Introduction

Extracellular vesicles (EVs) derived from mesenchymal stem cells (MSCs) have drawn increasing attention as a cell-free therapeutic modality in regenerative medicine due to their capacity to recapitulate many paracrine functions of MSCs [1,2]. These nanosized vesicles (30–200 nm), enclosed by a lipid bilayer, transport diverse bioactive molecules—including proteins, lipids, mRNAs, and microRNAs (miRNAs)—that mediate intercellular communication and tissue repair processes [3].

In dermatology, MSC-EVs have been explored as potential agents for reversing cutaneous aging and stimulating hair follicle regeneration [4]. Skin aging is characterized by diminished fibroblast function, reduced collagen synthesis, accumulation of oxidative damage, and compromised extracellular matrix architecture [5]. Concurrently, hair loss disorders such as androgenetic alopecia involve follicular miniaturization, dermal papilla cell senescence, and dysregulated hair cycling—particularly a shortened anagen phase [6]. Conventional treatments, including topical minoxidil and oral finasteride, often yield variable outcomes and may present adverse effects without addressing underlying regenerative deficits [7].

MSC-EVs present a promising alternative, offering regenerative cues without the ethical, immunologic, or tumorigenic risks associated with cell-based therapies [8]. Preclinical evidence has shown that MSC-EVs promote fibroblast proliferation, upregulate type I collagen, suppress matrix metalloproteinases (MMPs), and activate dermal papilla cells [4]. Nevertheless, the biological efficacy of EVs is closely tied to their cellular origin and production conditions, which influence their molecular cargo and therapeutic potency.

Our previous in vitro research established a biochemical stimulation method—referred to as the “Bio-Pulsed” approach—that enhances the bioactivity of EVs derived from avian MSCs. This strategy increased the regenerative functions of skin fibroblasts and dermal papilla cells, suggesting the presence of enriched miRNAs and protein mediators [9]. Although these initial findings support the potential of Bio-Pulsed EVs, further clinical validation is essential.

MiRNAs within EVs are recognized as central regulators of skin and follicular regeneration, modulating inflammation, pigmentation, matrix remodeling, and cellular proliferation. Specific miRNAs—such as miR-21-5p, miR-27b-3p, miR-181a-5p, and miR-199a-5p—have been identified as key effectors in these processes [10,11,12]. Yet, comprehensive profiling and validation of these miRNAs in clinically relevant EV preparations remain limited.

To address this, the present study investigates the clinical performance of Bio-Pulsed avian MSC-derived sEVs through two human trials targeting scalp and facial skin. These trials aim to evaluate the safety and efficacy of topical Bio-Pulsed sEV formulations using quantitative biophysical measurements and molecular analysis, offering insights into their translational potential for regenerative dermatology.

## 2. Materials and Methods

### 2.1. Reagents

Avian mesenchymal stem cell-derived small extracellular vesicles (AMSC-sEVs) were produced following our previously established protocol [9]. Briefly, AMSCs were isolated from *Gallus gallus* domesticus embryos and cultured in Dulbecco’s modified Eagle’s medium (DMEM) supplemented with a high glucose concentration (4.5 g/L) (Thermo Fisher Scientific Inc., Waltham, MA, USA) and 5% FBS. For Bio-Pulsed sEV production, cells were stimulated with 10 μg/mL *Polygonum multiflorum* extract for 72 h in a chemically defined medium. The resulting conditioned medium was collected and processed using tangential flow filtration (TFF, 100 kDa MWCO), followed by size exclusion chromatography. sEVs were then lyophilized and stored at −20 °C. Control (naïve) AMSC-sEVs were prepared using the same method but without Bio-Pulsed stimulation. Both Bio-Pulsed and naïve AMSC-sEVs were characterized in accordance with MISEV2023 guidelines as previously described [9,13].

For clinical use, Bio-Pulsed AMSC-sEVs were formulated into two topical products (ExoGiov^®^ ampoule and essence, Emosoxe Biotech International L.L.C., New Castle, DE, USA), as detailed in Section 2.4.

### 2.2. miRNA Sequencing and Bioinformatic Analysis

To evaluate the miRNA cargo of naïve and Bio-Pulsed AMSC-sEVs, total RNA was extracted using the qEV RNA Extraction Kit (IZON Science, Christchurch, New Zealand) following the manufacturer’s instructions. RNA quality and integrity were assessed using the Agilent 2100 Bioanalyzer with the Small RNA kit (Agilent Technologies, Santa Clara, CA, USA). Small RNA libraries were constructed using the QIAseq miRNA Library Kit (Qiagen, Hilden, Germany), which incorporates unique molecular indices (UMIs) during reverse transcription and PCR amplification for improved quantification accuracy. Sequencing was performed on the NovaSeq 6000 platform (Illumina, San Diego, CA, USA). Raw reads were trimmed to remove adapter sequences and low-quality bases, then collapsed by UMI. Clean reads were mapped to miRBase, piRNABank, and Rfam databases for annotation. The expression matrix was normalized to counts per million (CPM) and log2-transformed using the EdgeR package (version 4.6.1) in R (version 4.3.1). Differentially expressed miRNAs (DEMs) were identified based on a threshold of |log2 fold change| ≥ 1 and *p* < 0.05.

### 2.3. miRNA Quantitative Real-Time PCR Analysis

Total RNA was extracted from naïve and Bio-Pulsed AMSC-sEVs using the Total RNA Purification Kit (Norgen Biotek Corp., Thorold, ON, Canada) according to the manufacturer’s protocol. RNA concentration and purity were measured using the Qubit™ RNA HS Assay Kit (Thermo Fisher Scientific, Waltham, MA, USA). Reverse transcription was performed using the TOOLS miRNA RT Kit (BIOTOOLS Co., Ltd., New Taipei City, Taiwan). Quantitative real-time PCR was conducted using the TOOLS miRNA RT-qPCR Assay and TOOLS Easy 2× Probe qPCR Mix on the QuantStudio 12K Flex Real-Time PCR System (Applied Biosystems, Waltham, MA, USA). Specific TaqMan-based probes were used to quantify gga-miR-21-5p, gga-miR-22-3p, gga-miR-22-5p, and gga-miR-199a-5p. U6 snRNA and synthetic cel-miR-39-3p were used as internal controls. Relative expression was calculated using the 2^−ΔΔCt^ method. All reactions were run in triplicate. Statistical significance was evaluated using unpaired two-tailed Student’s *t*-tests, with *p* < 0.05 considered significant.

### 2.4. Clinical Product Application Protocol

Two topical formulations containing Bio-Pulsed AMSC-sEVs were evaluated in human clinical studies targeting scalp and facial skin.

In the hair regeneration trial, 30 participants (aged 30–65 years) received a 60-day topical treatment using ExoGiov^®^ Bio-Pulsed Exosome Ampoule (Emosoxe Biotech International L.L.C., New Castle, DE, USA), consisting of 9 blue bottles (10 × 10^9^ EVs/5 mL) and 26 silver bottles (4 × 10^9^ EVs/5 mL). On Day 1 of each treatment week, one blue bottle was applied after shampooing and massaged into the scalp for 2–3 min without rinsing. On Days 2–7, 2.5 mL of the silver bottle formulation was applied daily in the same manner. This regimen was maintained for 60 days. Participants were advised to avoid hair dyeing, perming, and direct scalp UV exposure during the trial.

Clinical assessments were conducted at Day 0, Day 30, and Day 60 under controlled conditions (25 ± 2 °C; 60 ± 5% RH), including the following:•Anagen/telogen (A/T) ratio and telogen follicle percentage (T%), using the Aram Huvis ASW Scalp Analyzer (Aram Huvis Co., Ltd., Seongnam, Republic of Korea) and Motic DM-1802 microscope (MoticEurope; Barcelona, Spain).•Hair shedding via a standardized 60-stroke comb test, conducted by trained technicians under controlled conditions as previously described [14].•Hair density analysis using ImageJ software (version 1.54j) (National Institutes of Health; Bethesda, MD, USA), which processed and analyzed grayscale intensity from standardized trichoscopic photographs.•Subjective evaluations via an 8-item Likert scale questionnaire at Days 30 and 60, assessing hair condition, shedding, texture, and density (Appendix A).

In the skin rejuvenation trial, 30 female participants (aged 30–65 years) applied ExoGiov^®^ Bio-Pulsed Exosome Essence (30 mL; Emosoxe Biotech International L.L.C., New Castle, DE, USA) to the face twice daily for 28 days. The formulation contained lyophilized Bio-Pulsed avian MSC-derived sEVs at a concentration of 168 ppm (equivalent to 0.168 mg/mL), which corresponds to approximately 1.68 × 10^8^ Bio-Pulsed AMSC-sEVs per mL, based on a previously validated yield of 1 × 10^12^ Bio-Pulsed AMSC-sEVs per gram of freeze-dried powder. Thus, each 30 mL bottle delivered a total of approximately 5.04 × 10^9^ EVs. The formulation remained stable for up to 14 days after opening when stored sealed at 15–30 °C and protected from light.

Facial skin assessments included the following:•Wrinkle depth and pore size: Miravex Antera 3D (Miravex; Dublin, Ireland), which provides high-resolution 3D topographic surface analysis.•Skin firmness and collagen density: Cortex DermaLab Combo (Cortex Technology; Aalborg, Denmark), which combines suction-based elastometry and high-frequency ultrasound imaging (20 MHz) for real-time evaluation of dermal structural properties. Firmness values were derived from the skin’s retraction time post-suction, while collagen density was measured by echo intensity across dermal layers.•Melanin and erythema indices: CK Mexameter^®^ MX18 (Courage + Khazaka Electronic GmbH; Köln, Germany), which employs a tristimulus light source and two specific narrow-band wavelengths (red and green) to assess the concentration of melanin and hemoglobin in the epidermis and superficial dermis. Measurements were recorded in triplicate at each timepoint, and average values were analyzed.•Gloss and radiance: Zehntner ZGM1120 Glossmeter (Zehntner GmbH Testing Instruments; Sissach, Switzerland) evaluated at a 60° incident angle, providing objective gloss units for surface reflectivity.•UV spot and pigmentation: CK VisioFace RD (Courage + Khazaka Electronic GmbH; Köln, Germany) used for full-face imaging and quantification of UV-induced pigmentation and spot scores.•Subjective evaluations via an 8-item Likert scale questionnaire at Days 14 and 28 (Appendix A).

All procedures were reviewed and approved by the Institutional Review Board of National Cheng Kung University Hospital (NCKU HREC; Approval No. 113-182-2). Informed consent was obtained from all participants. Participant compliance and adverse events were monitored throughout. Instrument settings, imaging conditions, and analytical software were standardized to ensure consistency and reproducibility across assessments.

The complete excipient composition and concentrations of the three final topical formulations (Essence, Blue Ampoule, and Silver Ampoule) are provided in Appendix A. All excipients were selected based on biocompatibility, stability, moisturizing effect, and preservative efficacy to support extracellular vesicle integrity and topical delivery.

### 2.5. Statistical Analysis

All statistical analyses were conducted using GraphPad Prism version 9.0 (GraphPad Software Inc., San Diego, CA, USA). Data are presented as mean ± standard deviation (SD). Pre- and post-treatment comparisons were analyzed using paired, two-tailed Student’s *t*-tests. A *p*-value less than 0.05 was considered statistically significant.

### 2.6. Study Design and Inclusion Criteria

This study was designed as a prospective, single-arm, open-label clinical trial comprising two independent cohorts: (1) individuals with self-perceived scalp hair thinning and (2) women exhibiting clinical signs of facial skin aging.

For the hair regeneration cohort, 30 healthy male and female participants aged 30 to 65 years were enrolled. Inclusion criteria included mild to moderate hair thinning without active dermatological conditions affecting the scalp. Exclusion criteria included recent use (within 6 months) of topical or systemic hair loss treatments such as minoxidil or finasteride, hormone therapy, or recent scalp procedures.

For the facial skin cohort, 30 female participants aged 30 to 65 years were recruited, all presenting with visible signs of aging such as wrinkles, enlarged pores, or hyperpigmentation. Participants were instructed to avoid using other active skincare products during the study.

All study procedures were conducted in accordance with the International Council for Harmonisation Good Clinical Practice (ICH-GCP) guidelines and approved by the Institutional Review Board of National Cheng Kung University Hospital (Approval No. 113-182-2). Written informed consent was obtained from all participants prior to enrollment.

### 2.7. Use of Generative AI in Study Preparation

No generative artificial intelligence (GenAI) tools were used in the conception, design, data generation, statistical analysis, or scientific interpretation of this study. Generative AI (ChatGPT, powered by GPT-4o, OpenAI) was employed solely for language refinement purposes, including the editing of grammar, sentence structure, clarity, and formatting. The tool is available at: https://chat.openai.com (accessed on 15 May 2025). The use of GenAI did not influence the scientific content, experimental outcomes, or conclusions presented in this manuscript. All study findings and interpretations remain the sole intellectual contributions of the authors.

## 3. Results

### 3.1. Molecular Characterization of Bio-Pulsed AMSC-sEVs

To elucidate the molecular components underlying the observed regenerative effects, we performed small RNA sequencing and qRT-PCR to analyze the miRNA content of Bio-Pulsed AMSC-sEVs in comparison to naïve controls. Next-generation sequencing (NGS) revealed a significant enrichment of key miRNAs in the Bio-Pulsed group, including miR-21-5p, miR-22-3p, miR-27b-3p, miR-125b-5p, miR-181a-5p, miR-199a-5p, miR-122-5p, and miR-34a-5p (Table 1). These miRNAs are known to be involved in cellular processes related to dermal repair, pigmentation control, and follicular cycling. qRT-PCR validation confirmed upregulation trends for miR-21-5p, miR-22-3p, miR-199a-5p, and miR-122-5p, supporting the reliability of sequencing data (Figure 1).

### 3.2. Hair Regeneration Outcomes

A total of 30 participants completed the 60-day hair regeneration trial.

#### 3.2.1. Anagen-to-Telogen (A/T) Ratio

The A/T ratio, an established index of follicular activity, increased from 1.32 ± 0.40 at baseline to 1.62 ± 0.44 on Day 30 and to 1.97 ± 0.73 on Day 60. Statistical analysis using one-way ANOVA with Tukey’s post hoc test revealed that the increase from baseline to Day 60 was significant (mean difference = 0.6433; 95% confidence interval (CI): 0.3071 to 0.9796; *p* < 0.0001), as was the change from Day 30 to Day 60 (mean difference = 0.3503; 95% CI: 0.0141 to 0.6866; *p* = 0.0392). No significant difference was observed between baseline and Day 30 (*p* = 0.1004; 95% CI: −0.0433 to 0.6293). This 48.7% increase indicates a shift toward anagen-phase hair, suggesting activation of dormant follicles (Figure 2A).

#### 3.2.2. Percentage of Inactive (Telogen) Follicles (T%)

The proportion of follicles in the telogen phase decreased from 58% ± 11% at baseline to 52% ± 10% on Day 30 and to 47% ± 12% on Day 60. Tukey’s multiple comparisons revealed a statistically significant reduction from baseline to Day 60 (mean difference = 13.33; 95% CI: 6.903 to 19.76; *p* < 0.0001) and from Day 30 to Day 60 (mean difference = 7.333; 95% CI: 0.9032 to 13.76; *p* = 0.0213). The reduction from baseline to Day 30 was not statistically significant (*p* = 0.0726; 95% CI: −0.4302 to 12.43). Overall, this 23.1% decrease in telogen follicles over the treatment period indicates a favorable shift in follicular cycling toward active (anagen) phases, supporting the efficacy of the Bio-Pulsed AMSC-sEV-based intervention (Figure 2B).

#### 3.2.3. Hair Shedding and Density

Hair shedding, assessed via the 60-stroke comb test, was reduced from 8.60 ± 1.95 hairs at baseline to 6.87 ± 1.62 at Day 30 and 4.60 ± 1.38 at Day 60, representing a 46.5% reduction (*p* < 0.001). The adjusted mean difference between baseline and Day 30 was −1.73 hairs, with a 95% CI: −2.79 to −0.68 (*p* = 0.0005). Between baseline and Day 60, the mean difference was −4.00 hairs, with a 95% CI: −5.05 to −2.95 (*p* < 0.0001). From Day 30 to Day 60, hair shedding was further reduced with a mean difference of −2.27 hairs (95% CI: −3.32 to −1.21, *p* < 0.0001), indicating sustained treatment effects over time (Figure 3A). Image-based hair density quantification, derived from grayscale image analysis, revealed an improvement in scalp coverage over the 60-day treatment period. The average grayscale value increased from 110.14 ± 11.25 at baseline to 115.33 ± 12.06 on Day 30 and further to 118.46 ± 12.30 on Day 60. Although the increase between baseline and Day 30 was not statistically significant (mean difference: +4.81, 95% CI: −2.39 to +12.02, *p* = 0.2544), a significant enhancement was observed between baseline and Day 60 (+8.33, 95% CI: +3.50 to +13.10, *p* = 0.00194). The difference between Day 30 and Day 60 was not statistically significant (mean difference: +3.51, 95% CI: −3.69 to +10.72, *p* = 0.4788). These findings suggest a gradual increase in follicular coverage consistent with improved hair density over time (Figure 3B).

#### 3.2.4. Subjective Satisfaction

Participant-reported outcomes collected via structured questionnaires indicated high satisfaction levels. At Day 60, over 85% of subjects reported visible improvements in hair strength, volume, and scalp condition, with no adverse effects documented (Table 2).

### 3.3. Skin Rejuvenation Outcomes

All 30 participants in the facial trial completed the 28-day treatment course.

#### 3.3.1. Facial Wrinkle Depth and Skin Firmness

Wrinkle depth decreased progressively over the 28-day period, with a significant reduction observed at Day 14 (mean difference: −3.601, 95% CI: −6.930 to −0.2718, *p* = 0.0288) and Day 28 (−7.343, 95% CI: −10.67 to −4.013, *p* < 0.0001), corresponding to a 7.5% reduction from baseline (Figure 4A). Concurrently, skin firmness measured by elastometry increased by 14.0%, with significant improvements observed at Day 14 (mean difference: +11.11, 95% CI: +5.05 to + 17.18, *p* < 0.0001) and Day 28 (+14.20, 95% CI: +8.14 to +20.27, *p* < 0.0001) (Figure 4B), indicating improved dermal elasticity.

#### 3.3.2. Collagen Density and Pore Size

Collagen density, evaluated via high-frequency ultrasound, increased by 18.2% at Day 28 compared to baseline (mean difference: +18.74, 95% CI: +13.37 to +24.11, *p* < 0.0001) (Figure 5A). Pore area, evaluated using 3D topographic analysis, showed a significant reduction of 8.5% at Day 28 compared to baseline (mean difference: −11.43, 95% CI: −14.37 to −8.495, *p* < 0.0001) (Figure 5B), suggesting enhanced dermal matrix structure.

#### 3.3.3. Skin Tone, Pigmentation, Gloss, and Erythema

The melanin index, measured using the CK Mexameter, decreased by 6.5% at Day 28 compared to baseline (mean difference: −14.50, 95% CI: −17.76 to −11.23, *p* < 0.0001) (Figure 6A), suggesting significantly improved skin brightness. Skin gloss, assessed using the Zehntner Glossmeter at a 60° angle, increased by 12.3% from baseline to Day 28 (mean difference: +12.71, 95% CI: +7.44 to +17.97, *p* < 0.0001) (Figure 6B), indicating enhanced surface reflectivity.

Pigmented spots were quantified using the Antera 3D imaging system. The combined number and area of visible pigmentation decreased by 5.3% at Day 28 compared to baseline (mean difference: −5.36, 95% CI: −7.25 to −3.48, *p* < 0.0001) (Figure 6C), reflecting improved tone uniformity. UV-induced pigmentation, evaluated using the CK VisioFace RD, also showed a 5.3% reduction in average spot score at Day 28 compared to baseline (mean difference: 4.73, 95% CI: 2.91 to 6.55, *p* < 0.0001), indicating significant photoprotective and depigmenting effects (Figure 6D). The erythema index, measured via Antera 3D, showed a 3.9% reduction from baseline after 28 days of application (mean difference: 3.65, 95% CI: 0.67 to 6.63, *p* = 0.0101) (Figure 6E), indicating a mild but statistically significant soothing effect.

#### 3.3.4. Soothing Response to Irritation

To assess the soothing potential of the formulation, an erythema induction model was employed using 1% vanillyl butyl ether (Hot-Flux™) on the inner forearm. Following a 30 min occlusive application, test areas were treated with the Bio-Pulsed Essence or left untreated as a control. Redness levels were quantified using a Minolta Chromameter by measuring a-value changes.

At 60 min post-application, the test area treated with the Bio-Pulsed sEVs exhibited a significant 14.2% reduction in redness from immediate post-irritation values (95% CI: 10.31 to 17.31, *p* < 0.0001), compared to a 9.4% reduction in the untreated control site (95% CI: 6.12 to 12.60, *p* < 0.0001). This corresponds to a net soothing improvement of 51%, confirming a rapid and statistically significant anti-inflammatory effect of the Bio-Pulsed AMSC-sEV formulation (Figure 7).

#### 3.3.5. Consumer Perception

Subjective evaluation was conducted using a structured 5-point Likert scale questionnaire covering ten cosmetic parameters. After 28 days of product use, over 80% of participants reported satisfaction scores of ≥4.0 across most categories. Reported improvements included reduced fine lines, enhanced skin elasticity, improved brightness, and decreased pigmentation. No adverse reactions or discomfort were reported during the study period (Table 3).

## 4. Discussion

This study provides one of the first structured clinical evaluations of Bio-Pulsed avian mesenchymal stem cell-derived extracellular vesicles (Bio-Pulsed MSC-EVs) in esthetic dermatology. The dual application—targeting both hair regeneration and facial skin rejuvenation—yielded notable improvements in quantitative biophysical outcomes, including an increased anagen/telogen (A/T) hair ratio, reduced hair shedding, enhanced skin firmness, elevated collagen density, and diminished wrinkle depth. These clinical results are consistent with regenerative effects previously observed in preclinical models [9]. In that study, Bio-Pulsed AMSC-sEVs significantly suppressed LPS-induced upregulation of proinflammatory cytokines (IL-1β, IL-6, and TNF-α) in human skin fibroblasts under inflammatory challenge conditions. These findings demonstrated not only the absence of cytotoxic or proinflammatory effects but also the potential anti-inflammatory capacity of the Bio-Pulsed AMSC-sEVs. Such evidence provided a biologically sound and ethically justified rationale for subsequent clinical translation without redundant cytotoxicity assays. Furthermore, the applied concentration in the present trial (168 ppm, ~5.04 × 10^10^ particles per 30 mL bottle) was derived from this prior study, where particle concentrations ranging from 1 × 10^9^ to 1 × 10^11^/mL elicited favorable biological responses—including enhanced fibroblast proliferation, accelerated wound closure, and immunomodulation—without adverse cellular effects. The chosen dosage (~1.68 × 10^9^ particles/mL upon application) therefore resides within this effective, non-toxic range, reinforcing both the safety and mechanistic plausibility of the current clinical application.

A central feature of this study is the use of Bio-Pulsed priming technology, in which parent MSCs are biochemically stimulated using botanical compounds such as *Polygonum multiflorum* extract. This process is designed to optimize EV yield and functional bioactivity by enriching the vesicular cargo with regenerative components, including specific miRNAs and growth factor-associated proteins.

The regenerative activity of Bio-Pulsed AMSC-sEVs appears to be mediated, in part, by their elevated levels of functional miRNAs. Comparative analysis via NGS and qRT-PCR revealed significant upregulation of key miRNAs in Bio-Pulsed AMSC-sEVs relative to naïve controls (Table 1, Figure 1). Among them, miR-21-5p and miR-181a-5p are known to downregulate inflammatory signaling and enhance fibroblast-driven wound repair through modulation of TLR4 and TGF-β/Smad3 pathways, respectively [10,11]. Likewise, miR-199a-5p and miR-125b-5p have been implicated in pigmentation control and extracellular matrix remodeling via suppression of MITF and TYR [11]. The presence and qRT-PCR validation of these miRNAs in Bio-Pulsed AMSC-sEVs support their mechanistic role in the observed therapeutic effects.

In the hair regeneration trial, a 48.7% increase in the anagen/telogen (A/T) hair ratio and a 23.1% reduction in telogen (inactive) follicles (T%) were observed over the 60-day treatment period (Figure 2). These findings suggest enhanced follicular cycling and greater root anchoring, corroborating prior in vitro results demonstrating the ability of MSC-EVs to activate dermal papilla cells [9,12]. Our results also align with previous clinical observations by Gentile (2019), who reported improved hair density in androgenetic alopecia using autologous adipose-derived stem cell micrografts without adverse effects [15]. Moreover, a systematic review by Mao et al. (2023) supports the efficacy of cell-based interventions, including EVs, in promoting hair thickness and follicle regeneration [16].

The 46.5% decrease in hair shedding (Figure 3A) and the 7.6% increase in visual hair density (Figure 3B) further confirm the beneficial effects of Bio-Pulsed AMSC-sEVs on scalp hair parameters. These changes are comparable to, or in some cases exceed, those seen with conventional treatments such as minoxidil or finasteride, yet without the commonly reported side effects [7]. Subjective evaluation findings supported the objective improvements, with over 90% of participants reporting enhanced scalp condition, reduced shedding, and noticeable new hair growth (Table 2; Appendix A). Together, these results are consistent with the proposed mechanism that EVs positively influence the follicular microenvironment and extend the duration of the anagen phase [6,8].

In the facial skin trial, objective measurements revealed a 7.5% reduction in wrinkle depth and a 14.0% increase in skin firmness after 28 days of use (Figure 4). These improvements are likely attributable to enhanced collagen synthesis mediated by miRNAs, notably miR-181a-5p and miR-34a-5p, both of which are known to stimulate fibroblast function and extracellular matrix production [11,12]. Supporting this, collagen density increased by 18.2% (Figure 5A), and pore size decreased by 8.5% (Figure 5B), indicating measurable dermal and epidermal structural remodeling.

Skin tone-related parameters also showed measurable improvement. A 6.5% decrease in the melanin index and a 5.3% reduction in UV-induced pigmentation spots were observed (Figure 6A,D), likely mediated by miRNAs such as miR-199a-5p and miR-27b-3p that regulate melanogenesis. Additionally, a 12.3% increase in skin gloss and a 3.9% reduction in erythema (Figure 6B,E) suggest enhanced barrier function and a reduction in subclinical inflammation. These observations align with prior studies demonstrating the anti-aging and dermal remodeling capacities of MSC-derived EVs. A 2025 review further supports their role in promoting collagen production, mitigating oxidative stress, and stimulating epidermal regeneration [17].

The anti-inflammatory effects of Bio-Pulsed AMSC-sEVs were validated in an irritation model, where treated areas exhibited a 14.2% reduction in redness versus 9.4% in untreated controls, reflecting a 51% net improvement (Figure 7). This response is consistent with the immunomodulatory actions of miR-21 and miR-181a-5p [10] and suggests potential applications in sensitive or post-procedure skin care.

Consumer perception data mirrored objective improvements. Structured survey responses showed that ≥80% of participants rated product performance at 4 or higher across most parameters, including skin appearance, texture, and comfort (Table 3; Appendix A). These findings indicate that biophysical improvements were accompanied by perceptible cosmetic benefits.

Finally, while this study confirms the safety and clinical efficacy of topically applied Bio-Pulsed AMSC-EVs, broader clinical adoption of EV-based therapies requires addressing key translational challenges. These include the development of scalable bioreactor systems, robust quality control of vesicle identity and cargo, and validated methods for storage and delivery [18]. Moreover, the harmonization of international regulatory frameworks remains critical to ensure consistency, safety, and therapeutic reliability in EV-based product development [19]. Meeting these demands will be essential for advancing Bio-Pulsed MSC-EVs into widespread clinical use in regenerative and esthetic dermatology.

Strengths of this study include the evaluation of a novel cell-free therapeutic modality—Bio-Pulsed AMSC-sEVs—across two prospective clinical trials targeting hair and skin regeneration. Comprehensive assessments were conducted using both objective biophysical instrumentation and subjective questionnaires. Additionally, regenerative miRNAs with plausible biological relevance were identified, a favorable safety profile was confirmed with no adverse events, and transparency was ensured through detailed reporting and Appendix A.

Nonetheless, limitations should be acknowledged. First, the absence of both a placebo control and an active comparator group (e.g., topical minoxidil for hair regeneration) may limit direct interpretation of the treatment’s relative efficacy. This single-arm, open-label design was chosen intentionally to evaluate safety and feasibility in a first-in-human setting. While comparisons to historical or published benchmarks provide contextual insights, they cannot fully substitute for internally controlled trials. Future randomized, double-blind studies incorporating placebo and active comparator arms are currently being planned to validate and benchmark the observed therapeutic benefits. Furthermore, the subjective nature of questionnaire-based assessments introduces potential bias related to participant expectations. Efforts such as standardized instruction, consistent application protocols, and objective instrumental analyses were employed to mitigate this, yet observer-independent designs will be prioritized in future investigations.

Taken together, the present findings support the notion that biochemical priming through the Bio-Pulsed approach enhances the regenerative potential of AMSC-sEVs. Compared with other stem cell- or EV-based cosmetic interventions, this strategy offers a standardized, stable, and bioactive formulation amenable to topical administration [4,8]. Further validation through long-term, placebo-controlled studies and in vivo mechanistic investigations will be important to reinforce clinical recommendations and confirm sustained safety over time.

## 5. Conclusions

This study presents clinical evidence supporting the efficacy and safety of Bio-Pulsed AMSC-sEVs in promoting hair follicle activation and facial skin rejuvenation. Through two independent, prospective human trials, topical application of Bio-Pulsed AMSC-sEVs significantly improved objective parameters, including anagen/telogen hair ratio, hair density, wrinkle depth, collagen density, and skin pigmentation. These outcomes were further supported by mechanistic insights from next-generation sequencing and qPCR analyses, which revealed enrichment of regenerative microRNAs in Bio-Pulsed AMSC-sEVs, particularly those involved in inflammation modulation, melanogenesis, and extracellular matrix remodeling.

The non-invasive and cell-free nature of this approach offers a promising alternative to traditional regenerative therapies. Notably, the use of botanical priming to enhance vesicular cargo may provide a standardized strategy to optimize EV-based treatments. While the results are encouraging, further randomized controlled trials with larger cohorts and longer follow-up periods are warranted to validate these findings and explore broader clinical applications. These data collectively support the potential of Bio-Pulsed AMSC-sEVs as a novel, topically applicable therapeutic in esthetic and regenerative dermatology.

## Figures and Tables

**Figure 1 cimb-47-00539-f001:**
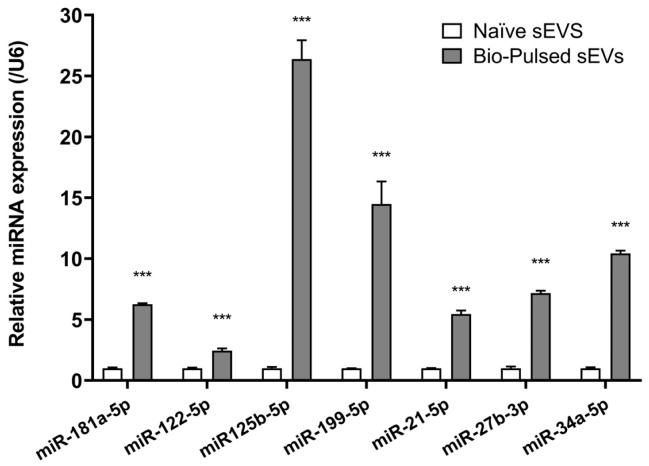
miRNA validation in Bio-Pulsed AMSC-sEVs. qRT-PCR validation of differentially expressed miRNAs. Fold-change expression levels of key regenerative miRNAs confirmed NGS trends, particularly in miR-21-5p, miR-22-3p, and miR-199a-5p (N = 3; data are expressed as the mean ± standard deviation; *** *p* < 0.001, compared with the naïve group).

**Figure 2 cimb-47-00539-f002:**
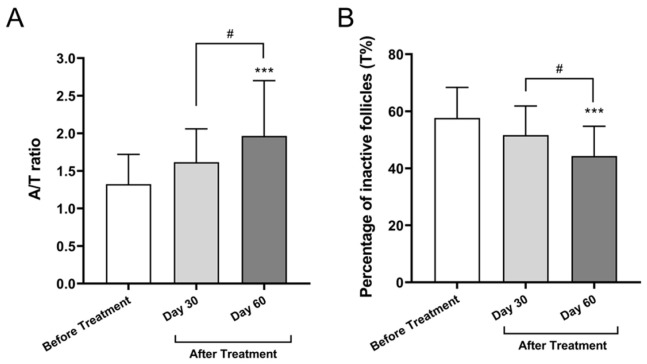
Changes in follicular growth phase indicators after treatment with Bio-Pulsed AMSC-sEVs: (**A**) Anagen-to-telogen (A/T) hair ratio measured at baseline, day 30, and day 60 using the photo-trichogram (PTC) method. (**B**) Percentage of telogen-phase (inactive) follicles (T%) measured at the same timepoints. Hair samples were collected from a 1.5 cm^2^ scalp area and analyzed using the ASW Scalp Analyzer. Data are presented as mean ± SD from 30 participants (*** *p* < 0.001, compared with before treatment; # *p* < 0.05, compared with Day 30).

**Figure 3 cimb-47-00539-f003:**
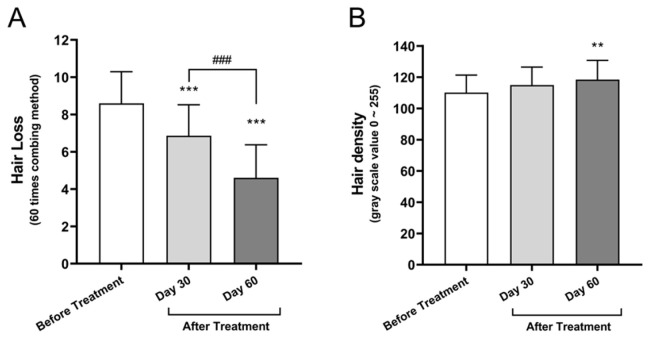
Assessment of hair shedding and density following Bio-Pulsed AMSC-sEVs treatment: (**A**) Hair shedding was measured using the standardized 60-stroke combing test. Participants combed their scalp 60 times within 30 s using a flat-tooth comb, and the number of shed hairs was recorded at baseline, day 30, and day 60. (**B**) Hair density was quantified using grayscale image analysis. Scalp images of the vertex region were binarized using IrfanView (version 4.70, released June 2024)and ImageJ software, and the average grayscale values (0 = black, 255 = white) were calculated to reflect hair coverage. Data are presented as mean ± SD from 30 participants (** *p* < 0.01, *** *p* < 0.001, compared with before treatment; ### *p* < 0.001, compared with Day 30).

**Figure 4 cimb-47-00539-f004:**
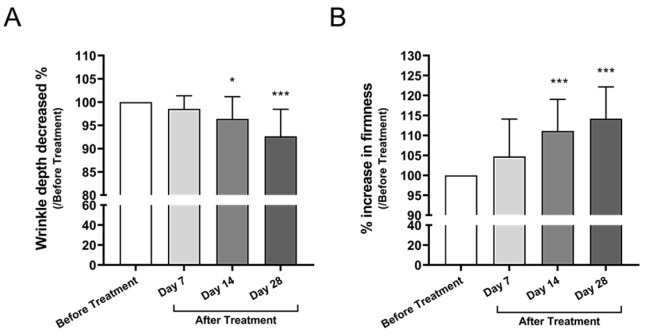
Improvement in facial wrinkle depth and skin firmness following Bio-Pulsed AMSC-sEVs treatment. (**A**) Wrinkle depth was measured using the Antera 3D imaging system at Days 0, 7, 14, and 28. Data are presented as percentage reduction relative to baseline. (**B**) Skin firmness was assessed by measuring Young’s modulus (E-values) using the DermaLab Combo elastometry device. E-values obtained at Days 7, 14, and 28 were normalized to baseline and expressed as percentage increase. Data are presented as mean ± SD from 20 participants (* *p* < 0.05, *** *p* < 0.001, compared with before treatment).

**Figure 5 cimb-47-00539-f005:**
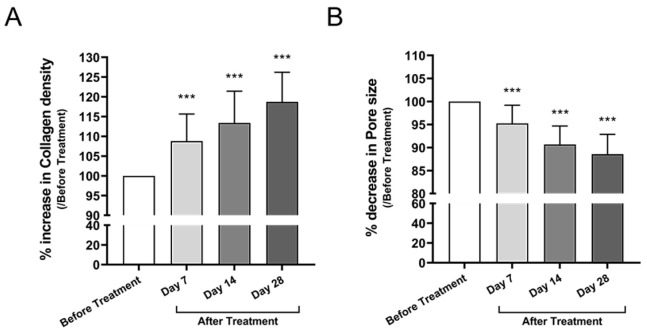
Changes in collagen density and pore size after treatment with Bio-Pulsed AMSC-sEVs. (**A**) Collagen density was assessed using the DermaLab Combo ultrasound system and expressed as a percentage increase relative to baseline at Days 7, 14, and 28. (**B**) Pore size was measured using the Antera 3D imaging system and expressed as percentage reduction from baseline over the same time course. Data are presented as mean ± SD from 20 participants (*** *p* < 0.001, compared with before treatment).

**Figure 6 cimb-47-00539-f006:**
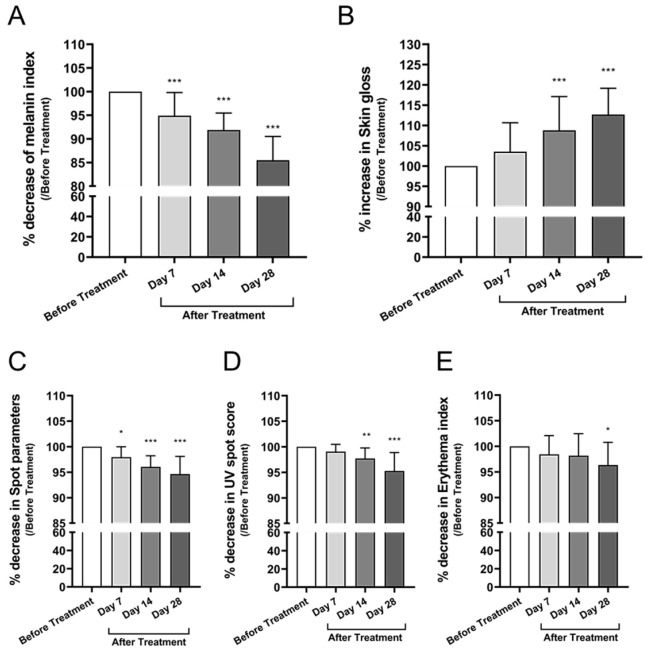
Changes in skin tone, pigmentation, gloss, and redness after treatment with Bio-Pulsed AMSC-sEVs. (**A**) Melanin index measured using the CK Mexameter. (**B**) Skin gloss evaluated by the Zehntner Glossmeter (60° angle). (**C**) Pigmented spot count assessed using Antera 3D imaging. (**D**) UV spot index determined by the VisioFace RD imaging system. (**E**) Erythema index measured using Antera 3D. Data are presented as mean ± SD from 20 participants (* *p* < 0.05, ** *p* < 0.01, *** *p* < 0.001, compared with before treatment).

**Figure 7 cimb-47-00539-f007:**
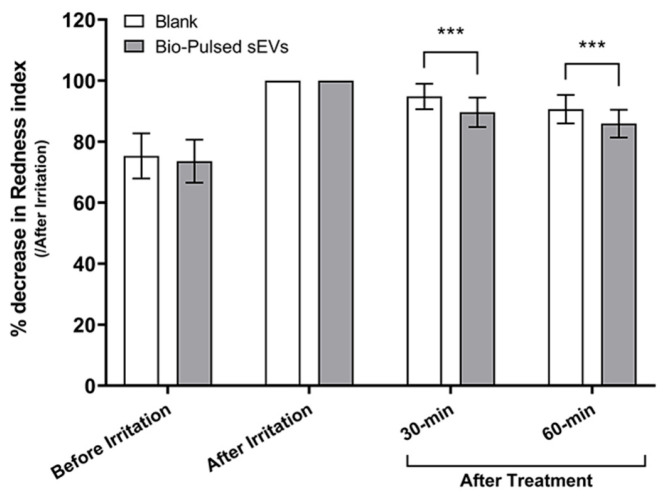
Soothing effect of Bio-Pulsed Exosome formulation on irritation-induced erythema. Erythema was induced using Hot-Flux™ (distributed by Corum Inc., Taipei, Taiwan) occlusive patches on the inner forearms of 30 subjects. The red color parameter (a-value) was measured at baseline, post-irritation, and after 30 and 60 min of topical application using a Minolta Chromameter (Konica Minolta Sensing, Tokyo, Japan). Data are shown as percentage reduction from peak redness. Bio-Pulsed Exosome-treated areas showed significantly greater redness reduction compared to untreated controls. Data are presented as mean ± SD from 30 participants (*** *p* < 0.001, compared with blank).

**Table 1 cimb-47-00539-t001:** The differential expressed miRNAs (DEMs) in naïve sEVs and Bio-Pulsed AMSC-sEVs.

DEMs	Reads of miRNAs		Log2FC	Regulation	*p*-Value
	Naïve AMSC-sEVs	Bio-Pulsed AMSC-sEVs			
gga-miR-181a-5p	15,304	98,713	2.6893	Up	<0.001
gga-miR-122-5p	1177	19,767	4.0699	Up	<0.001
gga-miR-125b-5p	25,640	1,759,371	6.1005	Up	<0.001
gga-miR-199-5p	925	59,526	6.0079	Up	<0.001
gga-miR-21-5p	210,645	1,423,493	3.9758	Up	<0.001
gga-miR-27b-3p	7915	68,897	3.1218	Up	<0.001
gga-miR-34a-5p	1091	191,776	7.4576	Up	<0.001

**Table 2 cimb-47-00539-t002:** Subjective evaluation of hair satisfaction at Day 30 and Day 60.

Evaluation Item	Mean Satisfaction Score			
	Day 30	Score ≥ 4.0 (%)	Day 60	Score ≥ 4.0 (%)
Reduced hair shedding	4.3 ± 0.66	90%	4.3 ± 0.61	93%
Increased hair volume	3.8 ± 0.83	77%	3.9 ± 0.69	87%
New hair growth	4.3 ± 0.66	90%	4.6 ± 0.49	100%
Strengthened hair roots	4.0 ± 0.61	83%	4.2 ± 0.73	83%
More active and healthier scalp	4.1 ± 0.51	93%	4.4 ± 0.57	97%
Reduced scalp sensitivity and itchiness	4.3 ± 0.66	90%	4.5 ± 0.57	97%
Reduced scalp oiliness	4.3 ± 0.61	93%	4.5 ± 0.57	97%
Hair appears fuller and more elastic	3.5 ± 0.63	60%	4.2 ± 0.90	90%

**Table 3 cimb-47-00539-t003:** Subjective evaluation of skin satisfaction after 14 and 28 days of using.

Evaluation Item	Mean Satisfaction Score			
	Day 14	Score ≥ 4.0 (%)	Day 28	Score ≥ 4.0 (%)
Reduce facial wrinkles and fine lines	3.8 ± 0.44	75%	4.2 ± 0.77	80%
Improved skin firmness	4.0 ± 0.56	85%	4.0 ± 0.21	85%
Improved skin elasticity	4.4 ± 0.49	100%	4.4 ± 0.49	100%
Improve enlarged pores	4.0 ± 0.86	65%	4.0 ± 0.76	70%
Brighten skin tone	4.0 ± 0.56	85%	4.4 ± 0.68	90%
Improve skin gloss	4.0 ± 0.60	80%	4.2 ± 0.77	80%
Improve uneven skin tone	3.6 ± 0.60	55%	4.4 ± 0.68	90%
Reduce facial spots	3.6 ± 0.51	55%	4.0 ± 0.76	70%
Reduce sensitive erythema on face	3.8 ± 0.64	65%	4.0 ± 0.51	85%
Overall skin quality improvement	4.2 ± 0.75	80%	4.2 ± 0.77	80%

## Data Availability

The clinical testing data generated and analyzed in this study are available from the corresponding author upon reasonable request. Full efficacy reports provided by SGS Taiwan Ltd. are not publicly posted due to file format limitations but can be made available for editorial or academic review upon request.

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
