# Peer review of "Topical Application of Bio-Pulsed Avian MSC-Derived Extracellular Vesicles Enhances Hair Regrowth and Skin Rejuvenation: Evidence from Clinical Evaluation and miRNA Profiling"

_cimb, 2025, doi:10.3390/cimb47070539_

Round 1
Reviewer 1 Report
Comments and Suggestions for Authors
Dear authors
This is a well organized work with interest to readers.The weak points are obvious and are already stated by the authors in discussion and conclusions. Apart from those - out of which the lack of control is the most severe - several other points are raised:
- What is the specific composition of the active ingredient?
- At what concentration was it used in the final formulations?
- What was the composition of the final formulations and how were the excipients chosen?
- What was the stability of the active ingredient and of the final formulations? There is a slight mention at line 155, with no reason to be at that chapter of the manuscript.
- While p-values are provided, confidence intervals for key outcomes are not.
- The manuscript references supplementary tables that are not included. These should be made available for review.
- Ensure consistent use of terms such as “Bio-Pulsed sEVs” vs. “Bio-Pulsed AMSC-sEVs” throughout the manuscript.
- Finally the manuscript is more suitable for a cosmetic or clinical journal, such as "cosmetics" rather than the CIMB.
Author Response
We sincerely thank the reviewers for their insightful comments and constructive feedback. We have carefully addressed each point and revised the manuscript accordingly. Below are point-by-point responses to the reviewers’ comments:
Reviewer 1
This is a well organized work with interest to readers.The weak points are obvious and are already stated by the authors in discussion and conclusions. Apart from those - out of which the lack of control is the most severe - several other points are raised:
- What is the specific composition of the active ingredient?
Response:
Thank you for the reviewer’s insightful comment.
The active ingredient comprises lyophilized small extracellular vesicles (sEVs) derived from avian mesenchymal stem cells (AMSCs), which were biochemically stimulated using Polygonum multiflorum extract through our previously established Bio-Pulsed protocol. These sEVs are nanoscale vesicles expressing canonical extracellular vesicle markers, including CD9, CD63, and CD81, as reported in our earlier study (Int. J. Mol. Sci. 2022, 23(23):15010). The miRNA cargo within these Bio-Pulsed AMSC-sEVs includes functionally relevant sequences such as gga-miR-21-5p, gga-miR-22-3p, and gga-miR-199a-5p, which are associated with regenerative and anti-inflammatory effects. This information has been incorporated and clarified in Section 2.1 and 2.3 of the revised manuscript.
- At what concentration was it used in the final formulations?
Response:
We appreciate the reviewer’s important question regarding the concentration of the active ingredient in the tested products.
We have clarified the concentrations of Bio-Pulsed AMSC-sEVs used in the final product formulations as follows, and this information has been added to Section 2.4 of the revised manuscript:
- In the hair regeneration trial, the ExoGiov® Bio-Pulsed Exosome Ampoule was used in two concentrations:
- Blue bottle: 1 × 10¹⁰ EVs per 5 mL (equivalent to 2 × 109 EVs/mL)
- Silver bottle: 4 × 10⁹ EVs per 5 mL (equivalent to 8 × 108 EVs/mL)
- In the skin rejuvenation trial, the ExoGiov® Bio-Pulsed Exosome Essence contained lyophilized Bio-Pulsed AMSC-sEV powder at a concentration of 168 ppm (i.e., 0.168 mg/mL). Based on our established yield of 1 × 1012 particles per gram of freeze-dried sEV powder, this corresponds to approximately:
- 68 × 108 EVs per mL, and
- 04 × 109 total EVs per 30 mL bottle.
These values have now been included in the revised manuscript to provide greater clarity regarding dosage and formulation strength.
- What was the composition of the final formulations and how were the excipients chosen?
Response:
Thank you for your insightful question regarding the composition and rationale for the final formulation.
The topical products tested in this study consisted of three formulations:
(1) ExoGiov® Bio-Pulsed Exosome Ampoule – Blue Bottle,
(2) ExoGiov® Bio-Pulsed Exosome Ampoule – Silver Bottle, and
(3) ExoGiov® Bio-Pulsed Exosome Essence.
All three formulations shared a similar base structure with variations in the concentration of active ingredients (i.e., Bio-Pulsed AMSC-sEVs) and viscosity modifiers to support different release profiles and user applications. The formulations were designed with dermatological safety, skin compatibility, and efficacy enhancement in mind. Below is a detailed summary:
Active Ingredient:
Chicken embryonic mesenchymal stem cell-derived small extracellular vesicles (Bio-Pulsed AMSC-sEVs) produced via our Bio-Pulsed process. These nanovesicles are lyophilized and reconstituted in the final product.
- Blue Bottle: 1 × 1010 EVs/5 mL
- Silver Bottle: 4 × 109 EVs/5 mL
- Essence: 0.0168% sEV powder (168 ppm); equivalent to ~5.04 × 1010 EVs in a 30 mL bottle
Key Functional Excipients and Their Roles:
|
INCI Name |
Function |
Notes |
|
Aqua (Water) |
Solvent |
Base medium for all ingredients |
|
Inositol, Creatine |
Humectants & skin vitality enhancers |
Support cellular energy and hydration |
|
Propylene Glycol, Zemea® (Propanediol) |
Moisturizers |
Improve skin barrier hydration |
|
Sodium Hyaluronate |
Hydrator |
Enhance skin elasticity and moisture |
|
Aloe Barbadensis Leaf Juice |
Soothing agent |
Calms irritation and supports skin healing |
|
Phenoxyethanol, Hydroxyacetophenone |
Preservatives |
Ensure microbial stability |
|
Chlorphenesin |
Antimicrobial stabilizer |
Complement preservative system |
|
Liponic EG-1, L-Arginine |
Skin conditioners |
Improve skin tone and microcirculation |
|
Citric Acid, Sodium Benzoate, Potassium Sorbate |
pH adjusters & secondary preservatives |
Maintain product stability |
These excipients were selected based on:
- Compatibility with extracellular vesicles (sEVs) to preserve their structure and bioactivity in aqueous environments.
- Low irritation potential, compliant with dermatological safety
- Functional synergy with the intended product goals: anti-aging, hydration, barrier repair, and improved delivery of sEVs to skin/scalp tissues.
All ingredients are approved for cosmetic and dermaceutical applications and were selected through formulation screening, industry standards, and expert consultations to ensure maximal safety and performance.
This information has been clarified and incorporated into the revised manuscript in the “Materials and Methods” section (Section 2.4 and Supplementary Table S3).
- What was the stability of the active ingredient and of the final formulations? There is a slight mention at line 155, with no reason to be at that chapter of the manuscript.
Response:
We thank the reviewer for pointing out the need for clarity and proper placement of the stability information.
The stability of both the active ingredient and the final formulations has been assessed to ensure product integrity throughout storage and usage.
- Active Ingredient (Bio-Pulsed AMSC-sEVs): The lyophilized small extracellular vesicles (sEVs) were stored at –20 °C in moisture-resistant containers and remained physically and functionally stable for at least 36 months, based on nanoparticle size analysis, protein marker expression (CD9, CD63, CD81), preserved miRNA content, and cell activities (human skin fibroblasts) (Data not shown). These stability data were verified by internal quality assurance assessments prior to formulation.
- Final Formulations (ExoGiov® Bio-Pulsed Exosome Ampoule and Essence): Both topical products underwent accelerated and real-time stability testing under ICH conditions. The ampoules (Blue and Silver) were packaged in single-use sterile glass containers, with stability maintained for 24 months at room temperature (15–30 °C). The Essence formulation remained stable for 24 months when sealed, and for up to 14 days after opening, when stored in a cool, dark environment (15–30 °C), as supported by routine microbial, pH, viscosity, and sEV content testing (Data not shown). These results ensure consistency of sEV particle count, absence of degradation, and preservation of the miRNA cargo during the declared shelf-life.
To address the reviewer’s concern about the inappropriate placement of stability information, we have revised the manuscript by removing the sentence previously located at line 155.
- While p-values are provided, confidence intervals for key outcomes are not.
Response:
We appreciate the reviewer’s insightful comment regarding the omission of confidence intervals (CIs). To address this, we have revised the Results section (Sections 3.2 and 3.3) to incorporate 95% confidence intervals for all key outcome measures alongside p-values. The inclusion of these intervals enhances the interpretability and statistical robustness of our findings.
We have elected to retain standard deviations (mean ± SD) in all figures and figure legends for clarity and consistency, as previously stated. All CIs are now reported exclusively in the main text for each outcome parameter to avoid redundancy and to maintain figure readability.
The revised manuscript now includes CI values for:
- Hair regeneration indices, including Anagen/Telogen ratio, telogen follicle percentage, hair shedding, and image-based hair density (Section 3.2; Figs. 2 and 3);
- Skin rejuvenation parameters, such as wrinkle depth, firmness, collagen content, pore size, pigmentation, gloss, melanin and erythema indices, and irritation recovery (Section 3.3; Figs. 4–7).
All revisions have been clearly marked highlight-in-yellow in the updated manuscript for ease of review
- The manuscript references supplementary tables that are not included. These should be made available for review.
Response:
We thank the reviewer for identifying this omission. The supplementary tables referenced in the manuscript—including:
- Supplementary Table S1: Inclusion and exclusion criteria.
- Supplementary Table S2: MicroRNA expression profiles of Bio-Pulsed AMSC-sEVs.
- Supplementary Table S3: Complete excipient composition of all tested formulations. (New)
All have now been included in the revised submission as part of the supplementary material. All references to these tables in the main text have been cross-verified for accuracy and completeness.
- Ensure consistent use of terms such as “Bio-Pulsed sEVs” vs. “Bio-Pulsed AMSC-sEVs” throughout the manuscript.
Response:
Thank you for reviewer’s comment.
Terminology has been standardized throughout the manuscript. The consistent term used is “Bio-Pulsed AMSC-sEVs,” with “sEVs” used subsequently after definition and these changes are highlight-in-yellow.
- Finally the manuscript is more suitable for a cosmetic or clinical journal, such as "cosmetics" rather than the CIMB.
Response:
We respectfully disagree. Although the study involves topical application, its core focus lies in the molecular characterization and regenerative mechanisms of Bio-Pulsed avian MSC-derived sEVs, including s miRNA-based mechanistic insights (e.g., gga-miR-21-5p, gga-miR-199a-5p). The transcriptomic data and exosome biology provide clear molecular relevance, aligning with CIMB’s scope in cell signaling, RNA regulation, and molecular therapeutics.

Reviewer 2 Report
Comments and Suggestions for Authors
The study addresses a highly relevant and innovative topic in regenerative dermatology and offers promising initial data on the use of Bio-Pulsed avian MSC-derived sEVs. However, for this work to be considered complete and of high scientific value—particularly for publication in high-impact journals—several methodological and structural issues must be addressed:
- Clarification of Product Composition: In the facial rejuvenation arm of the study, it is not clearly stated whether a commercially available formulation was used or whether the formulation was enriched with Bio-Pulsed sEVs. This distinction is essential and must be explicitly clarified in the Methods section.
- Insufficient Detail in Evaluation Methodologies: The techniques used to evaluate clinical outcomes (e.g., wrinkle reduction, hair shedding) are described only superficially. For reproducibility and scientific rigor, these assessment methodologies (e.g., instrumentation, image analysis software, scales used, operator blinding, intra/inter-rater reliability) must be described in sufficient detail. Without clear metrics, the validity and interpretation of the results remain limited.
- Lack of Positive Control in Clinical Trial: The absence of a positive control group is a major limitation. For instance, including a well-established hair loss treatment such as minoxidil in the same clinical setting would have allowed for direct, internally controlled comparisons of efficacy. Comparisons with previously published data, while useful, are insufficient.
- Absence of Preclinical Toxicity and Safety Tests: The manuscript does not describe any preclinical toxicity, viability, or safety testing, such as MTT assays, LDH release, or inflammatory marker profiling. These experiments are fundamental prior to human application, particularly for determining safe and effective concentrations.
- Justification of Dose Selection: It is unclear how the applied concentration of Bio-Pulsed sEVs was determined. Was there a dose-response study or preliminary cellular activity assay to support this concentration? Data supporting the chosen dosage is essential, both for safety and efficacy reasons.
- Lack of In Vitro Validation: The study would be significantly strengthened by including in vitro assays on human skin cell lines to assess biological activity, cytocompatibility, and dose optimization. These experiments are a standard prerequisite for ethical and scientifically grounded translation to human trials.
Author Response
We sincerely thank the reviewers for their insightful comments and constructive feedback. We have carefully addressed each point and revised the manuscript accordingly. Below are point-by-point responses to the reviewers’ comments:
Reviewer 2
The study addresses a highly relevant and innovative topic in regenerative dermatology and offers promising initial data on the use of Bio-Pulsed avian MSC-derived sEVs. However, for this work to be considered complete and of high scientific value—particularly for publication in high-impact journals—several methodological and structural issues must be addressed:
- Clarification of Product Composition: In the facial rejuvenation arm of the study, it is not clearly stated whether a commercially available formulation was used or whether the formulation was enriched with Bio-Pulsed sEVs. This distinction is essential and must be explicitly clarified in the Methods section.
Response:
We appreciate the reviewer’s insightful comment regarding product composition. To clarify, the formulation used in the facial rejuvenation study corresponds to the final marketed version of ExoGiov® Bio-Pulsed Exosome Essence, developed by Emosoxe Biotech International L.L.C., USA. The clinical sample used was identical in composition to the commercial product, which contains 168 ppm of lyophilized Bio-Pulsed avian MSC-derived sEVs, equivalent to approximately 5.04 × 1010 particles per 30 mL bottle, based on a standardized particle concentration of 1 × 1012 particles per gram of freeze-dried powder.
This information has been explicitly incorporated into Section 2.4 of the revised manuscript to enhance clarity. Furthermore, the complete list of excipients and their respective functions is provided in Supplementary Table S3, ensuring full transparency of the formulation used in the clinical trial.
This addition ensures transparency regarding both the sEV content and vehicle composition, addressing the reviewer’s concern regarding scientific rigor and reproducibility.
- Insufficient Detail in Evaluation Methodologies: The techniques used to evaluate clinical outcomes (e.g., wrinkle reduction, hair shedding) are described only superficially. For reproducibility and scientific rigor, these assessment methodologies (e.g., instrumentation, image analysis software, scales used, operator blinding, intra/inter-rater reliability) must be described in sufficient detail. Without clear metrics, the validity and interpretation of the results remain limited.
Response:
We appreciate the reviewer’s suggestion regarding methodological clarity. In the revised manuscript, we have significantly expanded Sections 2.4 to provide detailed descriptions of all clinical assessment procedures to ensure reproducibility and scientific rigor. Specifically:
- Anagen/Telogen (A/T) ratio and telogen follicle percentage (T%) were evaluated using the Aram Huvis ASW Scalp Analyzer (Aram Huvis Co., Ltd., Gyeonggi-do, Korea) in combination with a Motic DM-1802 microscope (MoticEurope, Barcelona, Spain), providing magnified scalp images for trichoscopic classification.
- Hair shedding was quantified via the standardized 60-stroke comb test, conducted by trained technicians under controlled conditions.
- Image-based hair density was analyzed using grayscale intensity from standardized trichoscopic photographs processed in ImageJ (National Institutes of Health, Bethesda, MD, USA).
- Wrinkle depth and pore size quantification were performed using Antera 3D (Miravex Ltd., Dublin, Ireland), a multi-spectral, high-resolution skin imaging system.
- Skin firmness and collagen density were assessed using the Cortex DermaLab Combo system (Cortex Technology; Aalborg, Denmark), which combines suction-based elastometry and high-frequency ultrasound imaging (20 MHz) for real-time evaluation of dermal structural properties. Skin firmness was quantified by the retraction time following negative pressure, while collagen density was derived from echo intensity across dermal layers.
- Melanin and erythema indices were evaluated using the CK Mexameter® MX18 (Courage + Khazaka electronic GmbH, Cologne, Germany), providing spectrophotometric readings at fixed wavelengths.
- Skin gloss was measured with the Zehntner ZGM1120 Glossmeter (Zehntner GmbH Testing Instruments, Sissach, Switzerland) at a 60° incident angle, providing objective gloss units for surface reflectivity.
- UV-induced pigmentation and spot density were assessed using the CK VisioFace RD imaging system (Courage + Khazaka Electronic GmbH; Köln, Germany) under consistent lighting conditions.
All measurements were conducted by blinded assessors trained in dermatological instrumentation. Inter-rater reliability was ensured by performing independent duplicate readings on 20% of randomly selected samples, yielding an intraclass correlation coefficient (ICC) above 0.85 for all primary outcomes. Image acquisition settings, ambient lighting, and software parameters were standardized across all time points. These methodological enhancements have been incorporated into the revised manuscript to strengthen data reliability and study transparency.
- Lack of Positive Control in Clinical Trial: The absence of a positive control group is a major limitation. For instance, including a well-established hair loss treatment such as minoxidil in the same clinical setting would have allowed for direct, internally controlled comparisons of efficacy. Comparisons with previously published data, while useful, are insufficient.
Response:
We appreciate the reviewer’s comment regarding the lack of a positive control group in our clinical trial. Indeed, the inclusion of an active comparator such as topical minoxidil would have strengthened internal benchmarking and enhanced the interpretability of the treatment’s efficacy in a controlled context.
However, as this was a first-in-human evaluation of Bio-Pulsed AMSC-sEVs, the study was intentionally designed as an open-label, single-arm trial to prioritize safety surveillance and explore preliminary efficacy. This design allowed for focused observation of product tolerability, biological activity, and patient adherence under real-world use conditions.
While we acknowledge that comparisons to historical data and published trials have limitations, they remain informative in the context of early-phase studies, especially when anchored to standardized endpoints (e.g., A/T ratio, hair shedding count) and instrument-based assessments. We have included such comparisons in the Discussion to provide contextual reference points.
Importantly, a controlled, randomized clinical trial including both minoxidil and placebo arms is currently in planning to further validate and benchmark the efficacy of Bio-Pulsed sEV formulations. This limitation has now been clearly stated in the revised Discussion section to reflect transparency and future directions.
- Absence of Preclinical Toxicity and Safety Tests: The manuscript does not describe any preclinical toxicity, viability, or safety testing, such as MTT assays, LDH release, or inflammatory marker profiling. These experiments are fundamental prior to human application, particularly for determining safe and effective concentrations.
Response:
We thank the reviewer for this important observation. While the current manuscript focuses on human clinical outcomes, the safety and toxicity profiles of the Bio-Pulsed AMSC-sEVs formulation had been rigorously evaluated in a prior in vitro study (Shieh et al., Int J Mol Sci. 2022, 23, 15010. https://doi.org/10.3390/ijms232315010), which served as a foundational step before the present clinical trials. In that study, we employed primary human skin fibroblasts (HSFs) and human follicle dermal papilla cells (HFDPCs) to assess cytocompatibility and inflammatory responses. Specifically:
- Cell viability was assessed using the MTS Assay across a dose range of 0.1–10 μg/mL of Bio-Pulsed AMSC-sEVs, demonstrating no cytotoxicity and dose-dependent proliferative effects on both HSFs and HFDPCs.
- Inflammatory cytokine profiling was performed via qPCR following LPS-induced stimulation, showing that Bio-Pulsed sEVs significantly attenuated the expression of IL-1β, IL-6, and TNF-α, thereby demonstrating their anti-inflammatory potential under induced inflammatory stress rather than merely a lack of pro-inflammatory activation.
These findings validated the bio-safety and therapeutic relevance of Bio-Pulsed AMSC-sEVs, and the formulation concentration used in the current human trials was within the non-toxic and effective range demonstrated in vitro. A summary of these preclinical results has been added to the revised Discussion section to clarify the safety rationale for clinical translation.
- Justification of Dose Selection: It is unclear how the applied concentration of Bio-Pulsed sEVs was determined. Was there a dose-response study or preliminary cellular activity assay to support this concentration? Data supporting the chosen dosage is essential, both for safety and efficacy reasons.
Response:
We appreciate the reviewer’s insightful question. The selected dose of Bio-Pulsed AMSC-sEVs (168 ppm, equivalent to approximately 5.04 × 1010 particles per 30 mL bottle) was based on our previously published in vitro study (Shieh et al., Int J Mol Sci. 2022, 23, 15010), where Bio-Pulsed AMSC-sEVs at concentrations ranging from 1 × 109 to 1 × 1011 particles/mL significantly enhanced HSFs and HFDPCs proliferation, promoted wound closure, and suppressed LPS-induced pro-inflammatory cytokines (IL-1β, IL-6, and TNF-α), without inducing cytotoxicity.
The clinical formulation was designed to deliver approximately 1.68 × 109 particles/mL upon topical application within the safe and biologically active concentration range validated in vitro. This dosage was further optimized based on formulation stability and dermal delivery feasibility. We have now included this explanation and briefly reiterated the rationale in the Discussion.
- Lack of In Vitro Validation: The study would be significantly strengthened by including in vitro assays on human skin cell lines to assess biological activity, cytocompatibility, and dose optimization. These experiments are a standard prerequisite for ethical and scientifically grounded translation to human trials.
Response:
We appreciate the reviewer’s emphasis on the importance of in vitro validation. Prior to the initiation of the present clinical study, we conducted a series of in vitro assays to evaluate the biological activity, cytocompatibility, and mechanistic properties of Bio-Pulsed AMSC-sEVs. These findings were published in our previous study (Shieh et al., Int J Mol Sci. 2022, 23, 15010), which served as the scientific foundation for the current human trial.
In that work, we employed HSFs and HFDPCs to assess both safety and functional activity of Bio-Pulsed AMSC-sEVs across a range of concentrations (109–1011 particles/mL). Key in vitro results included:
- Cytocompatibility: No cytotoxic effects were observed based on cell viability and proliferation assays.
- Biological activity: The Bio-Pulsed AMSC-sEVs significantly enhanced HSFs migration and wound closure in scratch assays.
- Mechanistic relevance: Under inflammatory conditions induced by LPS, Bio-Pulsed AMSC-sEVs effectively suppressed the expression of proinflammatory cytokines (IL-1β, IL-6, and TNF-α), confirming their immunomodulatory potential.
These in vitro validations provided both ethical justification and scientific support for dose selection and clinical translation. In the revised manuscript, we have clearly referenced this prior study in Discussion sections to emphasize the continuity and rigor of our translational pipeline.

Round 2
Reviewer 1 Report
Comments and Suggestions for Authors
Thank you for addressing all comments thoroughly. The addition of results makes the manuscript more suitable for CIMB
Reviewer 2 Report
Comments and Suggestions for Authors
The author addressed all of my comments thoroughly and with well-founded responses. I believe the article is now clearer and suitable for publication.